# Does Physical Activity during Alpine Vacations Increase Tourists’ Well-Being?

**DOI:** 10.3390/ijerph16101707

**Published:** 2019-05-15

**Authors:** Philipp Schlemmer, Cornelia Blank, Martin Schnitzer

**Affiliations:** 1Department of Sport Science, University of Innsbruck, 6020 Innsbruck, Austria; Martin.Schnitzer@uibk.ac.at; 2Institute for Sports Medicine, Alpine Medicine and Health Tourism, UMIT, 6060 Hall, Austria; cornelia.blank@umit.at

**Keywords:** sports tourism, health tourism, physical activity, vacation, well-being, alternative tourism, tourism strategies

## Abstract

Physical activities have been proven to have an impact on general well-being in everyday life; however, literature lacks an analysis of the effects of physical activities in vacation settings. Thus, the study aimed at assessing the impacts of physical activity on well-being during vacation by taking a longitudinal approach. We utilized a pre-post within-subject design (*n* = 101) by testing vacationers prior to, during, and after their vacation in an alpine environment. Therefore, a series of eight linear mixed model analyses of co-variance was performed. The results suggested that the duration of a vacation and the amount of physical activity have a positive impact on the components of well-being, which was expressed by changes in the activation, elation, excitement, and calmness subscales of the Mood Survey Scale. Demographic patterns did not reveal any influences. Physical activity might be a marker for well-being, which influences people’s everyday life and leisure time behavior by motivating them to engage in more physical activity. This research extends the existing literature by (1) proving the effects of vacations on well-being, (2) pointing out the effects of demographic predeterminations, and (3) gathering in-depth knowledge about the role of physical activity in changes to well-being.

## 1. Introduction

Vacation time is claimed to provide people with a valuable complement to everyday life [1,2]. Tourists consciously look forward to enhancing their well-being by engaging in tourism [3], and academic research has attempted to prove any impacts and relationships between vacations and well-being.

Lucas and Gohm [4] described well-being as an inhomogeneous phenomenon, which consists of independent components. Indeed, well-being is defined as ‘all of the various types of evaluations, both positive and negative, that people make of their lives’ [5] (p. 153). Hence, the holistic interaction of the absence of negative effects (unpleasant moods and emotions), the presence of positive effects (pleasant moods and emotions) as well as several other major components—like life satisfaction and satisfaction with other specific domains, such as leisure or family life—describe well-being [4,5]. Moreover, the domain of individuals’ spiritual interests and concerns are associated with well-being and subjective health [6,7]. According to Diener, Suh, Lucas, and Smith [8] these domains further include leisure, work, family, health, finances, the self and the individual’s social milieu.

There is an increasing number of campaigns, suggesting that the positive effects of any tourism-related experiential values may impact well-being positively [1]. However, well-being seems to be affected by vacation settings only in a minimal way [9,10,11]. Blank et al. [12], for example, showed that the implementation of short-term vacation (four nights) from the job highly affects perceived well-being—regardless of the vacation setting. The control group, who spent the vacation at home, also profited from the short vacation, however, to a lesser extent. Overall, Blank et al. [12] found effects of a short vacation even 30 to 45 days after the vacation in both groups.

The vacation setting is not the only aspect that was examined by researchers. Also, demographic variables, such as age or gender, seem to impact well-being; yet, literature is ambiguous about the extent to which they may have an influence [4,8,9,10,11,13,14,15,16,17]. In this line of research, literature has shown that age and gender have impacts on and are related to well-being in a daily routine [17]; however, in a vacation setting these impacts and relationships have not yet been examined [10,12].

In summary, recreational activities appear to have a greater impact on well-being, both positively and negatively [12,13], compared to the setting and/or demographics. However, this has not yet been systematically researched so far. Thus, this information gives rise to the question of whether vacation settings in general, demographic variables and/or physical activities have an impact on well-being in a vacation context. In addition, to the best of our knowledge, no information is available about the lasting effects of active (tourists engaging in physical activities) versus passive (tourists not engaging in physical activities) vacation behavior on well-being, while the general effects of leisure and recreational activities on well-being have already been examined [18]. Consequently, the current study aims (1) to build information about the impact of vacation settings on well-being and the sustainability of those effects after a vacation, (2) to prove whether demographic variables have an effect on well-being, and (3) to determine the effects of tourists’ physical activity levels on their well-being in tourism settings.

## 2. Literature

### 2.1. Well-Being

Subjective well-being, hereinafter referred to as well-being, is a rather intangible phenomenon, and yet, Becker [19] tried to operationalize well-being and render it more tangible. A multidimensional model distinguishes a chronological view, divided into current and habitual well-being as well as physical and psychological well-being. Pleasant and positive feelings, moods, and perceptions—both physical and psychological—as well as the absence of any complaints represent actual experiences of well-being. On the one hand, the current condition of well-being characterizes a person in a situation [19] and, according to Steyer, et al. [20], is usually multivariate and in the background of consciousness. On the other hand, influencing variables of habitual well-being are considered as steady attributes, indicating cognitive processes connected with aggregated emotional perceptions.

The comprehension of well-being has diverged in recent decades. By extension, actual as well as habitual well-being can further be substantiated by psychological (actual and prevailing mood in humans) and physical (current as well as habitual consciousness of complaints and general physical awareness) well-being assessments [21]. In 1999, the evaluation of Diener et al. [8] suggested the further integration of domains and background variables, such as leisure, work, family, health, finances, the self, and the individual’s social milieu, which all influence actual and habitual well-being. A broader involvement of well-being domains in human behavior [22] is determined in the self-determination theory (SDT) [23], which considers fundamental components, such as the fulfilment of needs or self-actualization, for the optimal development of well-being [24]. The self-determination theory is a macro-theory of human behavior and well-being [23,25], which focuses on volitional or self-determined behavior in a complementary approach of hedonic concepts and the concept of eudaimonia. This theory considers human beings as active organisms aiming at psychological growth and development of a personal sense of well-being [23,25].

Emerging from health psychological interests, Abele-Brehm and Brehm [21] conceptualized the Mood Survey Scale to measure well-being as well as the effects of physical activity on well-being. Its dimensions are formed by the bipolar dimensions of evaluation (positive and negative) as well as arousal (positive and negative). Each quadrant arising from the underlying concept is extended by two further dimensions respectively. Activation and excitement represent affective states with high arousal, whereas fatigue (negative valence) and calmness (positive valence) tag low arousal. Furthermore, anger and depression (both negative valence), contemplation (neutral valence), and elation (positive valence) characterize medium levels of arousal [21]. These considerations allow a useful comparison of the peculiarities of well-being due to different variables with higher values indicating a higher conformity to the subscale.

### 2.2. Well-Being after a Vacation

Well-being often constitutes the basis of studies measuring outcomes of interventions, activities or, for instance, vacation settings in a touristic context. The study by Gilbert and Abdullah [26] compared people taking a vacation with people taking no vacation in a twofold pre- and post-vacation approach. In contrast to the non-vacationers, people who went on vacation experienced higher life satisfaction and well-being levels prior to as well as after their actual vacation sojourn.

According to De Bloom et al. [27], vacations offer a more powerful opportunity to generally recover; however, De Bloom et al. [27] provided meta-analytical data indicating that levels of well-being manifested themselves differently in several studies, whereas a holistic consideration of all included studies reported small positive vacation effects with regard to well-being. One possible explanation for the mismatch between the current understandings of well-being and its influencing variables in scientific research could be the subjective effort justification of Festinger’s [28] cognitive dissonance theory. The effort justification refers to inconsistencies in personal cognitions, which cause uncomfortable psychological tensions, which means that people express themselves by assigning higher importance to one or more cognitions for re-establishing cognitive balance.

Furthermore, De Bloom et al. [27] showed that in most of the considered studies well-being decreased again after the vacation, constituting a small fade-out effect. However, only four studies applied post-vacation measurements, and no study included the impacts of certain physical activities during vacation. This relativizes the impact and keeps the effects of vacation activities unclear. Only one study included in the meta-analysis [29] described the effects of vacation experiences, revealing that negative work reflection during a vacation is associated with lower levels of well-being.

Further research related to the longitudinal impacts of vacations on health and well-being. An example is the study of De Bloom et al. [18], in which 96 Dutch workers self-reported their subjective health and well-being levels at five points in time, covering pre-vacation, vacation, and post-vacation periods. Health and well-being levels increased during the vacation, though post-vacation health and well-being levels mostly returned to base levels within one week after the vacation. These findings are supported by Strauss-Blasche et al. [11], who showed that health-related vacation outcomes are significantly affected, amongst others, by physical activity during vacation and individual vacation planning, respectively.

The obvious significance of a vacation for well-being and happiness is not given in all relevant academic research. In a cross-sectional study, Nawijn et al. [30] asked tourists in the Netherlands to complete a self-report questionnaire on well-being, whereby individual well-being levels were assessed at different stages during the vacation period. The results showed that the mood scores of vacationers were generally higher throughout the vacation time than the scores of non-vacationers, but lower at the beginning of the vacation period. Towards the end of the vacation, mood levels declined again and did not show any differences. Equivalent results were presented by Nawijn [10], who also showed marginal differences between vacationers and non-vacationers.

According to Blank et al. [12], short-term vacation from the job highly affects well-being, also 30 to 45 days after the vacation and regardless of the vacation setting as also vacation at home improved well-being (except for the variable perceived strain). In addition to the vacation setting, age, and gender have been stated to have an impact on well-being, and yet, scientific research has not been in unison about the extent to which they may have an influence [4,8,9,10,11,15,16,17]. In accordance, research has shown that age and gender affect and are related to well-being in a daily routine [16]; however, these impacts and associations have not been reflected in vacation-related literature. Thus, no impacts of age or gender on well-being or relationships between age or gender and well-being due to vacation setting were presented [9,11].

In spite of the current findings, researchers still consider demographic parameters, negative incidents and the choice of activities during a vacation as well as physical activities or the quality of sleep to influence tourists’ actual mental state [8,9,12]. Moreover, Su et al. [31] suggested enhancing tourists’ perceptions and experiences, postulating that they constitute an important step toward greater well-being.

As an example, the length of a vacation or longer periods of leisure time are widely believed to give rise to greater impacts on well-being than a shorter vacation [1,32,33]. To date, most studies have found no reliable evidence to suggest a clear relation between vacation length and positive vacation effects on well-being [34,35,36]. An exception, however, is provided by Blank et al. [37], who proved positive as well as sustainable impacts of short, health-related vacations (four overnight stays) on well-being even 45 days after the vacation. Taking account of the ambiguity of research—including single effects of certain parameters (demographic data)—led to the following hypotheses:

**H1:** 
*Well-being levels increase due to a vacation in the period between arrival and departure.*


**H2:** 
*Well-being levels do not decrease after a vacation in the period between departure and two to three weeks after departure.*


**H3:** 
*The impacts of vacation on well-being are different in males and females.*


### 2.3. Effects of Physical Activity during Vacation

The results from previous research outlined above raise the question of whether different vacation patterns, such as active and passive vacation behavior, may have different impacts on well-being and additionally affect the duration and strength of vacation effects. De Bloom et al. [9] tried to answer the question of possible activity effects. Their analysis considered the amount of time for physical, social, and passive activities and revealed that engaging in physical, social, and passive activities during a vacation contributes to changes in well-being and health; more specifically: the more time vacationers spent on physical activity, the better the effects on health and well-being.

Other results were presented by De Bloom et al. [35], who stated that vacation activities and experiences were only weakly associated with improvements in health and well-being. In contrast to the results of De Bloom et al. [9], De Bloom et al. [35] showed that passive activities and pleasure derived from these activities as well as relaxation, control, and sleep were strongly correlated with health and well-being improvements especially during a vacation, but additionally, to some extent also afterwards. These results seem to be consistent with the general consensus that physical activity and sports are seen as important influencing and predicting variables for well-being [38]. More recent research, though, has highlighted positive as well as persistent impacts health-related or physical activities during single and multiple short vacations have on well-being and other health parameters [37].

Retrospectively, a closer look at specific academic research has neither revealed an explicit conclusion about the impact of physical activity during a vacation on well-being, nor provided information about the amount of physical activity during a vacation and its impacts. Thus, there is an obvious dissent in research about the impact of physical activity levels on tourists’ well-being and its sustainability, which in turn leads to the following hypothesis:

**H4:** 
*Changes in well-being differ depending on the amount of physical activity during a vacation.*


## 3. Materials and Methods

### 3.1. Study Design

The current study utilized a longitudinal, observational approach (see Figure 1) with a pretest–posttest design by surveying tourists staying at a four-star hotel in the Austrian Alps and using an online questionnaire completed on tablet computers. The tourists were surveyed on three different occasions (t1: moment of arrival, t2: moment of departure, t3: back at home two or three weeks after their vacation). At t1, demographic parameters, habitual well-being, information about the arrival patterns (means of transport and travelling time) as well as the length of stay were of interest. At t2, current well-being as well as the level of physical activity during the vacation and the intensity of the activities were assessed. At t3, current well-being and the levels of physical activity were surveyed again. The tourists who were interviewed at their arrival at the hotel were encouraged to participate in the following surveys; those who were interested in further participation received the link to the respective questionnaires by email and were asked to complete them on the date of departure and two weeks after returning home.

### 3.2. Procedure

The study was carried out during the period from February 2017 to June 2018 by researchers at the entrance of a four-star hotel in the Austrian Alps. Tourists were questioned on their arrival at the hotel in varying seasons and on varying days of the week to secure a preferably heterogenic sample. The research team informed the respondents about the background and purpose of the study and stayed with them to overcome possible dubieties during the completion of the questionnaire. The second and third surveys were conducted by email with respondents who had specified their email address at the first questioning. Due to the fact that the respondents detailed the length of their stay (arrival and departure date), the researchers were able to deliver the questionnaires individually on the date of departure and two to three weeks after the respondents had returned home. The study at hand was conducted according to the ‘ethical guidelines for surveys’ approved by the Institutional Review Board (IRB) of the Department of Sport Science as well as the Board for Ethical Issues of the University of Innsbruck.

### 3.3. Sample

In total, 431 persons participated throughout the period of the survey; 101 of them were successfully recruited to fully complete all three questionnaires. 54.5% of them were male (*n* = 55) and 45.5% were female (*n* = 46). Minimum and maximum ages were 16 years and 73 years, resulting in an average age of 43.0 (SD = 13.5) years. The average number of overnight stays was 4.2 (SD = 1.6) with a minimum of 2 and a maximum of 7 overnight stays.

### 3.4. Measurements

Information assessed in the first questionnaire included demographic data (gender, age, level of education, income, occupation, and origin) as well as the level of physical activity. Well-being was examined via the standardized Mood Survey Scale [21,39] in German and English; the scale consists of an adjective list of 40 items assigned to eight subscales (activation, anger, elation, excitement, contemplativeness, depression, calmness, and fatigue) in quintets. The scale was implemented by a five-point Likert response mode (‘not at all’ to ‘very’) concerning habitual (‘Generally, I feel…’) as well as actual (“At the moment, I feel…”) perspectives of well-being; for example, the activation subscale comprised the statements ‘Generally, I feel…*active*’ and ‘At the moment, I feel…*active*’. The aggregation of five items for each of the eight subscales (activation, elation, contemplation, calmness, fatigue, depression, anger, and excitement) led to scores ranging from 5 (lowest value) to 25 (highest value). Information on psychometric properties to convergent and divergent validity are provided by Abele–Brehm and Brehm [21,39], who stated the specified internal consistency of their subscales between α = 0.73 and α = 0.88 (Cronbach’s alpha).

The level of physical activity was assessed by using the Godin Leisure-Time Exercise Questionnaire [40] dividing respondents into three groups. The categorization was conducted via the proposed formula (1), resulting in the Godin Scale Score:Weekly leisure activity score = (9 × Strenuous) + (5 × Moderate) + (3 × Light)(1)

Instead of surveying every single activity and its duration, the Godin Leisure-Time Exercise Questionnaire [40] focuses on the levels of strenuous, moderate as well as light physical activities, which are also prescribed in the questionnaire. The values for strenuous, moderate and light activities are addressed by the question ‘How many times on average do you engage in the following kinds of exercise (strenuous exercise—heart beats rapidly; moderate exercise—not exhausting; light exercise —minimal effort) for more than 15 min during your free time in a typical 7-day period (a week)’. Furthermore, the description of the categories is extended by the addition of certain physical activities equivalent to the certain category. The values given for strenuous, moderate, and light activities are multiplied by 9, 5, and 3, respectively. Afterwards, the respondents were divided into three groups: insufficiently active/sedentary people (<14 units), moderately active people (14–23 units) and active people (>24 units). To receive a detailed picture of the level of activity during vacation, the focus of this research, we used the respondents’ answers to the Godin scale [40], which was included in the survey at the departure. At this point in time, we asked the respondents to retrospectively assess their physical activity during the vacation.

### 3.5. Statistical Analysis

All statistical analyses were carried out using SPSS v. 24.0 (IBM Statistics, Armonk, NY, USA). Relative to the research questions, a series of eight (well-being scales) linear mixed model analyses of co-variance (ANCOVA) were performed with repeated measurements with a 3 (points in time) × 3 (activity level) × 2 (gender) design and with age as an additional covariate. Core outcomes were the main effect on well-being over time and interaction effects between well-being and activity level as well as between well-being and gender. For non-significant interaction effects but significant main effects on well-being Sidak-corrected post-hoc tests were used to provide more details on the effects. In case of a significant interaction effect (either gender or activity level), post-hoc linear mixed models were performed separately for the respective groups. A significant time effect between the three survey times (prior to the vacation at the arrival, after the vacation at the departure and back home in everyday life two to three weeks after the departure) was considered as a change in well-being due to vacation. Significant interaction effects were considered as an effect of the activity level and/or gender respectively. The level of significance was set at *p* < 0.05 with the additional information about the effect sizes (eta squared). Unless otherwise stated, data are presented as mean values (MV) with standard deviation (SD) or relative frequencies (%).

## 4. Results

### 4.1. Socio-Demographic Profile of the Respondents

The questionnaires were completed by 101 visitors at three points in time. The respondents came from Germany (45.5%), Austria (34.7%), the Netherlands (5.9%), Switzerland (2.0%), and other countries (11.9%), e.g., Israel; and their average vacation lasted 4.2 (SD = 1.6) overnight stays. The largest proportion of participants had completed a university degree (37.6%) followed by the group of 17.8% of the respondents who had accomplished an apprenticeship and 15.8% of the respondents who had a secondary school certificate. More than half (63%) of the participants were employed, whilst 23.0% of them were self-employed. Nearly a third (28.1%) of the participants disclosed a monthly average net income of €2000 to €3000; high income earners (>€3000 a month) represented about another third (29.2%) of the respondents; however, a considerable group of nearly a fifth (18.8%) did not want to answer the question.

### 4.2. Effects of Vacation on Well-Being

Vacation duration had a significant impact on well-being. Age did not show any associations with the variables of interest; thus, only gender was considered in the presentation of the results. With regard to H1 (and H2), the results of the effect of a vacation on the well-being subscales are presented in Table 1. The statistical analysis showed significant main effects of a vacation on the subscales of activation, elation, excitement, as well as calmness. Consequently, H1 can partially be supported by the present results.

Figure 2 outlines a more detailed analysis of the significant main effects. In this regard, activation over time is characterized by significant differences between the points in time arrival and departure (*p* < 0.05) as well as arrival and back home (*p* < 0.05), representing a continuous increase in activation throughout the vacation timeline and in its aftermath.

The values for elation share the same characteristics with the values for activation, revealing significantly differing subscales between arrival and departure (*p* < 0.05) as well as arrival and back home (*p* < 0.05). This means the main effect is still measurable after the actual vacation time.

Post-hoc analysis for the subscale of excitement shows significant differences between the points in time arrival and departure (*p* < 0.05), arrival and back home (*p* < 0.05), as well as departure and back home (*p* < 0.05). As presented in Table 1, the values for excitement decrease during the actual vacation and slightly increase again after the vacation but remain significantly lower than the initial levels.

Another peculiarity of the statistical analysis is the significant main effect of the subscale of calmness. Further post-hoc analysis revealed significant differences between arrival and departure (*p* < 0.05) as well as arrival and back home (*p* < 0.05), representing rising values until departure, which remain static and do not fade in the subsequent period. Thus, based on these results, H2 can also be partially supported.

### 4.3. Effects of Gender and Physical Activity during Vacation on Well-Being

No interaction effects were found for well-being × gender (H3). All subscales showed neither a significant interaction effect for activity level × well-being nor for gender × well-being. In consequence, H3 cannot be supported by the existing results.

The level of physical activity during a vacation according to Godin [41] was 31.8 (SD = 26.6) with a minimum rating of 3.0 and a maximum rating of 119.0. Significant interaction effects were found only for the activation scale (*p* = 0.013; η² = 0.064); concerning activation, well-being developed differently over time depending on the tourists’ activity level. For the insufficiently active/sedentary group well-being showed a significantly higher value when back home again compared to the value at the arrival at the hotel (*p* < 0.05). For the active and moderately active groups well-being improved when comparing the arrival with the departure values (active: *p* < 0.001; moderately active: *p* < 0.05) as well as the arrival with the back-home values (active: *p* < 0.001; moderately active: *p* < 0.05). All other aspects of well-being are not dependent on the level of physical activity. A more detailed overview can be found in Table 1. However, according to the available results, H4 is partially supported.

## 5. Discussion

Vacation time is an important dimension in peoples’ lives, indicating several impacts on well-being [8,9,10,11,27,33,41,42,43,44].

Key findings indicate manifold impacts of vacation time and physical activity levels on well-being during and after a vacation. Especially, the subscales activation, elation, excitement, and calmness are favorably impacted by vacation with activation also displaying an interaction effect with physical activity. However, these findings are only the tip of the iceberg; they are confirmed by several findings indicating differences caused by the amount of physical activity.

According to the presented data, vacation settings in general have several significant effects on well-being, justifying the partial agreement of H1. The subscales of activation, elation, excitement, depression, and calmness are influenced significantly by the level of engagement in physical activity during vacation, providing evidence of a medium effect of vacation on well-being as well as sustainable effects beyond the vacation period. Thus, psycho-physiological impacts occur; the recovery from everyday life as a consequence of the discontinuation of several stressors (occupation, education, etc.) might explain the findings of the study in hand. Another way of explaining the outcomes could be the application of Hobfoll’s [45] approach, stating that ‘individuals strive to obtain, retain, protect, and foster those things they value’ [45] (p. 341). Sonnentag [43] provided the conclusion that further engagement in recreation (vacation, physical activities) could serve as resource investments, which might be followed by positive resource gains. The changes in the subscales of well-being might also be explained by the feeling of ‘escaping everyday life’ [46] or just the sheer fact of having no responsibilities (e.g., occupation) directly restricting life. The opportunity for and appreciation of recreation time as well as recreational activities are assumed to be important causal influencing parameters for long-term happiness [47]; this is supported by the findings of the current study, which show prolonged positive effects of vacation sojourns. The application of these considerations implicates that vacation time in general offers the potential to recover and further boost levels of well-being. Additionally, these benefits are strengthened by the fact that there are no vacation effects on the subscales of anger, contemplativeness, and fatigue. In relation to the overall vacation effects, considerable research [12,18,26,29] supports the findings of the study in hand.

Regarding H2, which is partially supported by the results, the study revealed significant vacation effects, which are still measurable two to three weeks after returning home. These findings disagree with the current understanding of the durability of vacation effects [11,18,27,30], although they support the findings of Blank et al. [37]. One possible explanation for the mismatch with the current consensus in scientific research is the subjective effort justification of Festinger’s [28] cognitive dissonance theory. In short, higher-than-expected expenditure on the vacation sojourn could trigger the compulsion to state better well-being levels in accordance with higher spending.

The general assumption of literature [8,10,12] about the effect of gender on the development of well-being is not reflected in the current study (H3). The demographic characteristics do not seem to play any role in the constitution of well-being during vacation and its follow-up period. Heo, Lee, McCormick, and Pedersen [16] argued that well-being is positively influenced by engagement in serious leisure activities when retired. Further contrary findings are reported by Diener et al. [8] as well as Lucas and Gohm [4] who highlighted the negative effect of decreasing life conditions on well-being but acknowledge the opposite, positive effect of the increased ability to regulate emotions aligned to rising age. The findings of Brajša-Žganec et al. [13] showed an even but not significant decline in well-being due to aging, which cannot be supported by the study in hand.

Moreover, it should be taken into account that staying in the four-star hotel in the Austrian Alps, where the respondents spent their vacation, requires a certain income or financial assets; a factor, which could explain the contrary findings of the study. Another explanation for the outcomes could be the importance of social interaction in recreational activities, leading to higher life satisfaction whilst furthermore influencing well-being levels [48]. However, the current outcomes are also in conflict with the findings of Sun, et al. [49], who showed decreasing well-being levels aligned with age and significantly lower values for women than for men in Eastern countries. The ‘How’s Life? 2017′ report published by the OECD [50] underlined these assumptions, showing regionally differing values for well-being due to age and gender. These differences may be explained by regional differences in the relevant populations. The current study mostly focused on German and Austrian vacationers and thus citizens of developed countries and a presumably privileged group with lower inequalities, spending their vacation in a four-star hotel.

The assumption about the crucial importance of physical activity during vacation for well-being (H4) is partially supported by the present study. The amount of physical activity during a vacation crucially influences well-being components such as activation, both during a vacation and in its aftermath, when the respondents have returned home. It is quite interesting that the values for activation are significantly higher in people who are more active during a vacation than in people who are less active. The general acknowledgement of physical activity in the composition of well-being has already been present in scientific research as Gauvin, et al. [51] as well as Ruseski et al. [37] referred to essential positive affective and predictive characteristics and mediating effects of sports and physical activity. These effects do not always occur on their own, but are mediated by other factors, such as duration of leisure time, social contacts and time in nature [14,40,52,53,54,55].

De Bloom et al. [35] revealed that vacationers spent about a quarter of their vacation time on physical activity, though did not associate any changes in well-being with the amount of physical activity during a vacation. On the contrary, Blank et al. [37] showed that instructed physical activity leads to higher impacts on well-being in the intervention group than in the control group without any indications of physical activity. Apparently, the results suggested that people who practice physical activity during vacation show generally higher values for the positive well-being subscales and generally lower values for the negative components of well-being; these findings go in line with Abele-Brehm and Brehm [21,39]. Nonetheless, one could also assume that the differences of the subscale of activation in relation to the amount of physical activity are characteristic for well-being and might be a marker that influences the everyday life behavior of less ambitious people in the context of physical activity in leisure time and motivates them to engage in more physical activity.

As an antecedent of increased well-being levels due to physical activity, SDT might bear an explanation [23]. This theory distinguishes an intrinsic and extrinsic type of motivation, regulating subjective behavior. People who are intrinsically motivated experience feelings of enjoyment or personal accomplishment. Teixeira et al. [22] stated that the engagement in recreational sports and exercise leads to enjoyment or to subjective challenges of participating in an activity. These considerations may lead to differences in the degree of well-being. Another interesting issue in this context was brought up by Niedermeier et al. [55]. They yielded the information that there is a difference between indoor and outdoor physical activity, which raises the question of whether physical activity itself or physical activity in combination with nature cause the changes in well-being. However, due to the fact that the study at hand did not apply a control group, this issue cannot be answered. Though, the health-related effects of physical activity are associated with both indoor and outdoor environments [56].

## 6. Implications

Vacation showed medium-sized effects on well-being, partially confirming the current literature. Conversely, the sustaining effects of physical activity on well-being after a vacation in relation to the amount of physical activity are a novelty. Well-being is a continuum, in a scientific as well as in a practical approach. Vacations combining recreation with physical activity might be noted as a resource for tourism destinations, tourism businesses (e.g., hotels), various parties involved (touristic supply), direct dependents on tourism (e.g., employees) or even new suppliers in tourism (e.g., e-bike rentals), whose primary strategies are to provide key attractions for tourists [57,58]. In this context, the orientation toward an activity-based tourism offers the opportunity for new and alternative tourism concepts. Building a deeper knowledge of the longitudinal development of well-being through physical activity highlights the possibility of taking a year-round economic approach to tourism, leading to greater security in tourism planning. Furthermore, promoting active vacation in alpine destinations would help to further utilize existing infrastructure (e.g., mountain railways for skiing, hiking, and biking), which could first enhance the productivity and subsequently the profitability of service providers in tourism. Seasonal workers could, likewise, benefit from a year-round approach with guaranteed employment. Moreover, tourists’ loyalty as well as their behavioral intentions to revisit a destination [59,60,61,62] might be influenced by well-being, supported by life satisfaction, which itself is directly affected by leisure-time engagement.

Consequently, recent trends in activity-based tourism, contingent on the concept of well-being, show great potential in the tourism and sports industry as well as in public health, highlighting an opportunity for tourism destination management organizations to develop corresponding strategies [63,64,65]. Research recommends the utilization of more resources for innovative product development in tourism to ensure further crucial development. These considerations not only touch upon accommodation providers, but also the structure of each particular destination. However, the realization of advancements in tourism requires the right in-depth communication to successfully reach out to the tourism consumer.

## 7. Limitations

The study in hand also has some limiting factors, for example, the fact that the actual point in time for completing the online questionnaire was a little blurred as the respondents did not always fill out the questionnaire on the day of departure, but a day before or after. A further limiting factor is the small remaining group of tourists who participated in the entire survey (*n* = 101), the necessity to complete three questionnaires, and the fact that researchers had to be on site to contact possible respondents obviously reduced the number of evaluable datasets. Another problem was the further communication between researcher and respondents via email, in which the majority of people stopped participating in the research project. Moreover, the limited regional examination of respondents and a possible restriction of the informative value of the current study stand alongside the conscious decision to take a single hotel approach in order to eliminate possible surrounding effects (facilities, features etc.). Furthermore, there is the possibility that participants might have been inclined to respond that they had been more active than they actually were, because their physical activity levels were indicated retrospectively and based on their own responses. Another limitation to be mentioned is a possible effect of the environment [55,56]. The study at hand did not apply a control group to assess whether being active in nature has an additional effect on well-being or whether the positive effect on well-being can be traced back to the environment in which the vacation took place rather than to the activity. Thus, the results of this study should not be generalized for all tourism contexts but only for those with a similar setting in alpine destinations. This juxtaposition clearly provides the basis for further discussion.

## 8. Conclusions

Against the background of current touristic developments, which indicate a rise in the recreational interests of tourists, offering physical activities could add a unique facet to a vacation destination. The findings suggest a mainly positive effect of a simple vacation sojourn on well-being; however, the amount of physical activity is a further distinguishing factor, leading to higher levels of well-being in combination with vacation time.

In spite of everything, the question must be asked as to whether there is a critical threshold for physical activity, which, if exceeded, negatively influences well-being. There is, however, no definitive understanding of the construct of well-being, which depends on numerous factors and therefore requires further insights so as to understand the complexity of well-being and its dependent or influencing variables. Hopefully, the present findings can be taken into account for further and future research in the field of well-being or, more precisely, the general effects of physical activities, the quantity of activities, and the effects of different and autonomous activities on well-being in vacation situations.

## Figures and Tables

**Figure 1 ijerph-16-01707-f001:**
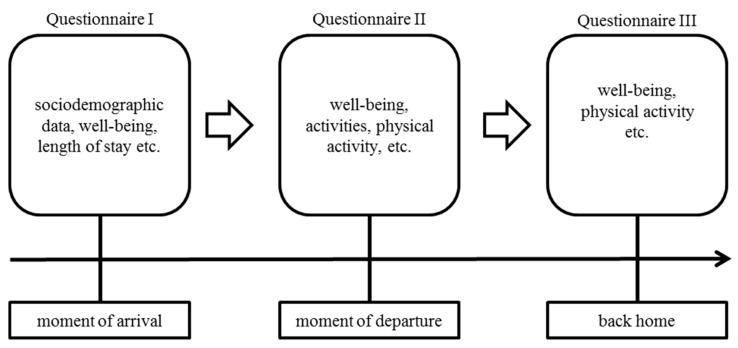
Study design, presenting the procedure of the longitudinal questionnaire approach.

**Figure 2 ijerph-16-01707-f002:**
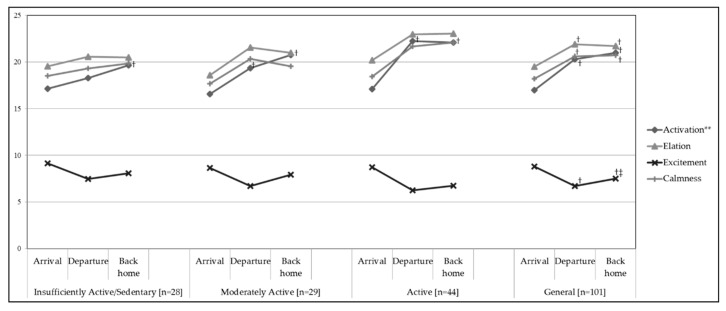
Graphic overview of significant subscales according to linear mixed model analysis of co-variance (ANCOVA) with repeated measurement, adjusted by activity level; ** significant group effects; † significant difference compared to arrival; ‡ significant difference compared to departure.

**Table 1 ijerph-16-01707-t001:** Overview of data according to linear mixed model analysis of co-variance (ANCOVA) with repeated measurement, adjusted by activity level. A = active; MA = moderately active; IAS = insufficiently active/sedentary; ^†^ significant difference compared to arrival; ^‡^ significant difference compared to departure; ^*^ significant effect.

Subscale	Group	*n*	Arrival	Departure	Back Home	Well-Being		Well-Being × Group		Well-Being × Gender	
			(MV ± SD)	(MV ± SD)	(MV ± SD)	*p*	η^2^	*p*	η^2^	*p*	η^2^
Activation	General	101	17.0 ± 4.3	20.3 ± 3.4 ^†^	21.0 ± 3.0 ^†^	<0.001 *	0.104	0.013 *	0.064	0.922	0.001
	A	44	17.1 ± 4.1	22.2 ± 2.1 ^†^	22.1 ± 1.9 ^†^						
	MA	29	16.6 ± 4.7	19.3 ± 4.0 ^†^	20.8 ± 3.5 ^†^						
	IAS	28	17.1 ± 4.4	18.3 ± 2.8	19.6 ± 3.3 ^†^						
Anger	General	101	6.4 ± 2.2	5.8 ± 1.9	5.9 ± 1.7	0.337	0.011	0.542	0.016	0.814	0.002
	A	44	6.5 ± 2.4	5.5 ± 1.0	5.5 ± 1.1						
	MA	29	6.3 ± 1.9	5.9 ± 2.6	5.8 ± 1.2						
	IAS	28	6.4 ± 2.2	6.0 ± 2.2	6.6 ± 2.5						
Elation	General	101	19.5 ± 3.3	21.9 ± 2.4 ^†^	21.7 ± 3.1 ^†^	<0.001 *	0.090	0.212	0.031	0.636	0.005
	A	44	20.2 ± 3.3	23.0 ± 1.6	23.0 ± 1.8						
	MA	29	18.6 ± 3.5	21.6 ± 2.8	21.0 ± 4.2						
	IAS	28	19.5 ± 2.7	20.6 ± 2.4	20.5 ± 2.9						
Excitement	General	101	8.8 ± 3.2	6.7 ± 2.3 ^†^	7.5 ± 2.1 ^†, ‡^	0.011 *	0.047	0.649	0.013	0.333	0.012
	A	44	8.7 ± 3.1	6.3 ± 2.0	6.8 ± 1.7						
	MA	29	8.7 ± 3.0	6.7 ± 2.2	7.9 ± 2.3						
	IAS	28	9.1 ± 3.5	7.5 ± 2.7	8.1 ± 2.3						
Contemplativeness	General	101	10.6 ± 3.7	10.0 ± 3.5	9.8 ± 3.8	0.954	0.001	0.525	0.004	0.666	0.004
	A	44	10.1 ± 3.8	9.5 ± 3.4	8.9 ± 3.3						
	MA	29	10.9 ± 3.3	10.3 ± 3.2	10.5 ± 3.8						
	IAS	28	11.0 ± 4.0	10.6 ± 4.0	10.5 ± 4.4						
Depression	General	101	6.5 ± 2.4	5.6 ± 1.2	6.1 ± 1.8	0.406	0.009	0.351	0.023	0.761	0.003
	A	44	6.5 ± 2.6	5.5 ± 1.0	5.6 ± 1.1						
	MA	29	6.4 ± 1.9	5.9 ± 1.7	6.5 ± 2.2						
	IAS	28	6.7 ± 2.5	5.5 ± 1.0	6.6 ± 2.0						
Calmness	General	101	18.2 ± 3.9	20.6 ± 3.0 ^†^	20.7 ± 3.2 ^†^	<0.001 *	0.089	0.116	0.038	0.857	0.002
	A	44	18.4 ± 3.9	21.7 ± 2.4	22.1 ± 1.7						
	MA	29	17.7 ± 4.3	20.3 ± 2.9	19.6 ± 3.8						
	IAS	28	18.5 ± 3.7	19.3 ± 3.5	19.8 ± 3.5						
Fatigue	General	101	9.2 ± 3.4	7.7 ± 3.0	7.0 ± 2.8	0.320	0.012	0.756	0.010	0.929	0.001
	A	44	8.7 ± 3.2	6.7 ± 2.0	6.1 ± 2.0						
	MA	29	9.6 ± 3.6	8.3 ± 3.6	7.1 ± 2.4						
	IAS	28	9.9 ± 3.5	8.8 ± 3.1	8.5 ± 3.4						

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
