# Peer review of "Does Physical Activity during Alpine Vacations Increase Tourists’ Well-Being?"

_ijerph, 2019, doi:10.3390/ijerph16101707_

Round 1
Reviewer 1 Report
This article analyzes the increase tourists well-being during alpine vacations through physical activities (PA).
That study present a high degree of novelty, since there is no other research or close in this field in the literature.....great for authors!!!!
I propose that the authors use the following style of abstract: Background, Methods, Results, Conclusions.
I think the Introduction is too short, the authors must detailed more this subsection. Also, the authors should add current research on this topic or close to it, see Blank C. (2018) et al. '' Short Vacation Improves Stress-Level and Well-Being in German-Speaking Middle-Managers-A Randomized Controlled Trial '' or others research.
Also, I think the authors must include in Introduction which is the novelty of this research.
Line 14 - The authors must detailed which was the interpretation value of the questionnaire of MSS??
I think a single table is not enough to present the results.....please add more data for this chapter. The authors have to show more results in this direction .... this also leads to an improvement of the article.
Reviewer 2 Report
This paper aims to analyse the impact of physical activity during Alpine vacation on tourists’ well-being. The topic is interesting as well-being issues are increasingly discussed, but so far they have not been widely studied in the context of tourism. However, revisions are needed to improve the manuscript before it will be acceptable in the journal. More detailed comments and suggestions follow:
Introduction
- p. 1, line 35: “The increasing number of campaigns suggesting that…” -> something wrong with the sentence (a verb is missing?)
- do the authors refer with the term ‘vacation’ to travelling/tourism, or does the term also include time spent at home on vacation? This is unclear also on p. 2-4 (chapters 2.2., 2.3.).
- one hypothesis is related to the effect of gender, but no information is given about the results of the previous studies (HOW gender has been suggested to impact well-being?), e.g. p. 1, lines 39-41
Materials and methods
- 3.3. Sample: e.g. “an average age of about 43.0 (13.5.) years” -> what does 13.5. mean (also 1.6. in the next sentence)?
- 3.4. Measurements: the study is based on Mood Survey Scale, but the content of the 40 items remains hidden as no examples of the items are given in the paper. Moreover, the publication of Abele-Brehm and Brehm is written in German, and thus, most readers of the journal are not able to read it to check the items. Therefore, items of all subscales should be shortly described in the paper so that the readers could get the idea of the survey questions.
- it also remains totally unclear how physical activity during a vacation was enquired in the survey. Moreover, in the results section there is no description about respondents’ physical activity during a vacation (how often, what activities, how exhausting etc.). Physical activity is one of the main themes of the paper, but it is impossible to get any idea of respondents’ physical activity during a vacation or the environment in which it took place (e.g. outdoors or indoors – did they go to a gym, or skiing or hiking or what did they do and how often?)
Results
- 4.2.: Was age included in the analyses (e.g. Table 1)?
- 4.3.: respondents’ physical activity DURING a vacation still remains totally unclear. There is information about different activity groups, but they do not tell anything about activity DURING a vacation if these questions concerned a TYPICAL week (not a vacation) as described in the methods section?
Discussion
- the role of the vacation environment should be discussed in the paper. There is a lot of research on the well-being effects of nature (e.g. Puhakka et al. included in the references), and probably also this research area in the Austrian Alps is located in the natural environment (no information about the area or tourists’ activities is given). Therefore, the authors should discuss is it really physical activity or natural environment which enhances tourists’ well-being; the information about the respondents’ activities is also needed in the paper. There are previous studies which have compared the well-being effects of physical activity indoors, outdoors in built environments and in nature, and found out the added value of nature to the known benefits of physical activity (e.g. Pasanen, Tyrväinen & Korpela 2014; doi: 10.1111/aphw.12031). This aspect should also be discussed in the limitations of the paper (the role of environment was not included in the analysis).
Limitations
- it should be considered that the information about the respondents’ physical activity was based on their own responses – people might be inclined to respond to be more active than they are in reality. Different kinds of accelerometers etc. could be used in future studies to measure physical activity during a vacation.
Reviewer 3 Report
This was a great and interesting paper to read. The authors go into sufficient depth with the argumentation for the relevance of their research, based on the existing literature. The modelling of well-being as well as its operationalisation into a research design was clearly descreibed and systematically implemented. The discussion also considered a number of potential limitations (e.g. psychological biases) and provided an internpretation of the results. The only areas of improvement I would propose would be to elaborate a bit more on the statistical and design scope limitations (i.e. specific form / segement of tourism, specific case of accommodation). Otherwise, a big 'well done' to the authors and all the best with thei publication
Reviewer 4 Report
I am sure you should add a reference to the article on 'sport tourism' by Malchrowicz-Mosko & Munsters, published in "Ido Movement for Culture" in 2018. There are also five interesting works written by Ivo Jirasek, published in the same Journal (See: www.imcjounal.com, open access).
For your scientific framework should be added something more about cultural anthropology, e.g.: Munsters W. & Melkert M. [eds.], Anthropology as a Driver for Tourism Research. Antwerpen: Garant.
Round 2
Reviewer 4 Report
Dear Authors,
Works written by Malchrowicz-Mosko and Munsters are really connected (more) with sport and cultural issues. However, Ivo Jirasek wrote some articles about psychological and humanistic aspects of tourism. Did you read it?
